# Role of Endogenous Lipopolysaccharides in Neurological Disorders

**DOI:** 10.3390/cells11244038

**Published:** 2022-12-14

**Authors:** Manjunath Kalyan, Ahmed Hediyal Tousif, Sharma Sonali, Chandrasekaran Vichitra, Tuladhar Sunanda, Sankar Simla Praveenraj, Bipul Ray, Vasavi Rakesh Gorantla, Wiramon Rungratanawanich, Arehally M. Mahalakshmi, M. Walid Qoronfleh, Tanya M. Monaghan, Byoung-Joon Song, Musthafa Mohamed Essa, Saravana Babu Chidambaram

**Affiliations:** 1Department of Pharmacology, JSS College of Pharmacy, JSS Academy of Higher Education & Research, Mysuru 570015, Karnataka, India; 2Centre for Experimental Pharmacology and Toxicology, Central Animal Facility, JSS Academy of Higher Education & Research, Mysuru 570015, Karnataka, India; 3Section of Molecular Pharmacology and Toxicology, Laboratory of Membrane Biochemistry and Biophysics, National Institute on Alcohol Abuse and Alcoholism, National Institutes of Health, Rockville, MD 20892, USA; 4Department of Anatomical sciences, School of Medicine, St. George’s University Grenada, West Indies FZ818, Grenada; 5Q3CG Research Institute (QRI), Research & Policy Division, 7227 Rachel Drive, Ypsilanti, MI 48917, USA; 621 Health Street, Consulting Services, 1 Christian Fields, London SW16 3JY, UK; 7National Institute for Health Research Nottingham Biomedical Research Centre, University of Nottingham, Nottingham NG7 2UH, UK; 8Nottingham Digestive Diseases Centre, School of Medicine, University of Nottingham, Nottingham NG7 2UH, UK; 9Department of Food Science and Nutrition, CAMS, Sultan Qaboos University, Muscat 123, Oman; 10Aging and Dementia Research Group, Sultan Qaboos University, Muscat 123, Oman

**Keywords:** lipopolysaccharide, endotoxemia, gut microbiota, gut–brain axis, neuroinflammation, neurodegeneration

## Abstract

Lipopolysaccharide (LPS) is a cell-wall immunostimulatory endotoxin component of Gram-negative bacteria. A growing body of evidence reveals that alterations in the bacterial composition of the intestinal microbiota (gut dysbiosis) disrupt host immune homeostasis and the intestinal barrier function. Microbial dysbiosis leads to a proinflammatory milieu and systemic endotoxemia, which contribute to the development of neurodegenerative diseases and metabolic disorders. Two important pathophysiological hallmarks of neurodegenerative diseases (NDDs) are oxidative/nitrative stress and inflammation, which can be initiated by elevated intestinal permeability, with increased abundance of pathobionts. These changes lead to excessive release of LPS and other bacterial products into blood, which in turn induce chronic systemic inflammation, which damages the blood–brain barrier (BBB). An impaired BBB allows the translocation of potentially harmful bacterial products, including LPS, and activated neutrophils/leucocytes into the brain, which results in neuroinflammation and apoptosis. Chronic neuroinflammation causes neuronal damage and synaptic loss, leading to memory impairment. LPS-induced inflammation causes inappropriate activation of microglia, astrocytes, and dendritic cells. Consequently, these alterations negatively affect mitochondrial function and lead to increases in oxidative/nitrative stress and neuronal senescence. These cellular changes in the brain give rise to specific clinical symptoms, such as impairment of locomotor function, muscle weakness, paralysis, learning deficits, and dementia. This review summarizes the contributing role of LPS in the development of neuroinflammation and neuronal cell death in various neurodegenerative diseases.

## 1. Introduction

The presence of lipopolysaccharide (LPS) reflects the biochemical characterization of the outer cell wall of predominantly Gram-negative bacterial species, such as *E. coli, Yersinia pestis, Klebsiella pneumonia, Pseudomonas aeruginosa, Porphyromonas gingivalis, bifidobacterial species Chlamydia trachomatis* and *Francisella tularensis,* etc. [1], but it is absent in Gram-positive bacteria. Primarily, LPS is made up of glycolipids and acts as an important membrane barrier between the cytosolic contents and the external environment of the bacteria. In the bacterial cell membrane, LPS is positioned towards the external environment, i.e., the outer leaflet of the bacterial membrane, whereas the phospholipids constitute the inner leaflet, i.e., towards the cytosolic contents, to reflect an asymmetric bilayer membrane [2], as shown in Figure 1. The susceptibility of Gram-negative bacteria to antibiotics and bile salts increases with the disruption of the LPS layer via a foray of phospholipids or loss of synthesis/transport [3]. Due to its higher molecular mass (>100 KDa), LPS is classified as a large oligosaccharide polymer. LPS glycolipid acts as an identification antigen marker for bacteriophages and the human immune system. LPS binding proteins (LBPs) are known to detach LPS from bacterial membranes and accelerate its binding to the pattern recognition receptor CD14 located on the surfaces of macrophages, monocytes, and dendrites [4] and toll-like receptors (TLRs), leading to stimulation of an inflammatory response.

The pathogenic roles of endogenous LPS are gaining much attention, as they are recognized to play a vital role in promoting neuroinflammation in a variety of neuropsychiatric diseases (e.g., depression, schizophrenia, anxiety, autism spectrum disorders, attention-deficit hyperactivity disorder, etc.) and neurodegenerative diseases (NDDs; e.g., Alzheimer’s disease (AD), Parkinson’s disease (PD), multiple sclerosis (MS), Amyotrophic lateral sclerosis (ALS), and stroke) [5,6,7]. Patients with neurological disorders develop gastrointestinal (GI) symptoms several years prior to the onset of the neuronal dysfunction itself and these GI symptoms include constipation, diarrhea, nausea, abdominal pain, bloating, vomiting, and altered appetite. These patients develop gut microbial dysbiosis, which is characterized by a disruption to the microbiome, resulting in an imbalance in the microbiota, such as changes in their functional composition/abundance and metabolic activities. LPS can be endogenously produced in the gut of human and mammalian species by the pathogenic Gram-negative microbes. Hence, gut microbial dysbiosis is a well-established causative factor for the release of excessive amounts of LPS from the pathogenic microbes (e.g., *E. coli*) [8,9]. LPS which leaks from the gut epithelial and mucosal lining enters the systemic circulation (termed as endotoxemia), interacts with the toll-like receptors (TLRs) on the immune cells, and triggers a cascade of pro-inflammatory reactions, leading to a central and peripheral cytokine storm. LPS-induced excessive cytokine production affects the immune homeostasis and promotes oxidative/nitrative stress in the gut, periphery, and nervous system via the gut–brain axis (GBA). Based on this connection, LPS is considered an immunotoxin and neurotoxin since it elicits chronic inflammation both peripherally and centrally. GBA is a bidirectional communication pathway where the brain influences the gut while signals from the gut are conveyed to the brain. The contributing role of LPS in NDDs is linked to its ability to increase proteinopathies such as aggregation and accumulation of amyloid-β, tau, α-synuclein, etc. [10,11]. Molecular studies have shown that LPS-mediated inflammatory reactions include impaired autophagy and elevated oxidative/nitrative stress, which is reflected by increased production of reactive oxygen species (ROS) and reactive nitrogen species (RNS), reduced activity of antioxidant enzymes (superoxide dismutases and glutathione peroxidase), dysfunctional endoplasmic reticulum and mitochondria with decreased efficiency of the respiratory chain. These changes ultimately result in cellular energy depletion, atrophy, and death, especially in neuronal cells. LPS-induced inflammatory mechanisms both in the peripheral and central immune cells as well as in the neurons and glial cells (e.g., microglia and astrocytes) have been actively studied in detail as targeting these inflammatory mechanisms to help prevent or delay the onset of several neurological disorders [12,13]. This review summarizes the contributing role of LPS in the etiopathogenesis and pathophysiology of several major NDDs.

## 2. LPS Biochemistry

### Kdo2-LipidA Biosynthesis

The complex structure of LPS and its biochemistry are described via the Raetz pathway [14,15]. LPS comprises of three distinctive layers: the first layer is the lipid A layer which acts as a membrane anchor, the second layer is the core oligosaccharide, and the third layer is the repeating O antigen polysaccharide, as illustrated in Figure 1. Lipid A is the biologically active portion of the LPS and plays a significant role in LPS-induced endotoxemia and tissue injury. It is the membrane anchor of LPS in Gram-negative bacteria. There are nine enzymes that catalyze the synthesis of Kdo2-lipid A [16]. The nine enzymes of this constitutive Kdo2-lipid A pathway and the single copy of the genes encoding them are preserved in most of Gram-negative bacteria. Of the nine enzymes, Lpx-A, -C, and -D are cytosolic, whereas Lpx-B and Lpx-H are peripheral membrane proteins. Lpx-K, Kdt-A, Lpx-L, and Lpx-M are integral inner membrane proteins.

The first step in this pathway is the fatty acylation of UDP-GlcNAc to UDP-3-O-GlcNAc. This reaction is catalyzed by Lpx-A. In *E. coli*, the acyltransferase LpxA requires an acyl carrier protein (ACP) as its donor substrate. Uridine diphosphate N-acetylglucosamine (UDP-3-O-GlcNAc) is further deacetylated by the zinc-dependent metalloamidase, LpxC, as the critical step in the lipid A biosynthetic pathway, to form UDP-3-O (R3 hydroxy myristoyl D-glucosamine). Following deacetylation, a second R-3-hydroxymyristate chain is further added by LpxD to generate UDP-2,3-diacyl-GlcN. The pyrophosphate linkage of this diacyl molecule is cleaved by LpxH to produce 2,3-diacyl-GlcN-1-phosphate (lipid X) and UMP. The cleaved lipid X condenses with UDP-2.3-diacyl GlcN in the presence of a glucosyltransferase, LpxB, to generate the β1,6-linked disaccharide. After this step, the last four steps are catalyzed by the integral membrane proteins, viz., LpxK, KdtA (WaaA), LpxL (HtrB), and LpxM (MsbB). LpxK phosphorylates at the 4′ position of the disaccharide, resulting in the formation of lipid IVA. Next, two Kdo residues are merged by the bifunctional enzyme KdtA. Finally, the Kdo2-lipid A biosynthetic pathway involves the addition of lauroyl and myristoyl groups to the glucosamine unit with the support of LpxL and LpxM, respectively. The latter enzymes LpxL and LpxM prefer acyl-ACP donors, however, the same can also be performed with acyl-coenzyme A.

The O antigen of the LPS is a polymer with variable oligosaccharide subunits. It is synthesized in two steps within the cytoplasmic membrane. In the first step, the repeating sugars are assembled by the stepwise addition of each sugar in the presence of bactoprenol-P, which is a lipid carrier. In the second step, the repeating sugars on the outer surface of the cytoplasmic membrane are polymerized by releasing one bactoprenol-P for each repeating sugar, which results in the formation of a repetitive chain [17]. The repeating O antigens are the most variable part of LPS, which imparts the antigen specificity properties. It determines the virulence potential of the bacteria and may help the bacteria to surpass the host’s defense mechanisms [18]. LPS is detected, even at picomolar concentrations, by host receptors such as CD14 and TLRs, which are present on the surface of macrophages and endothelial cells in the peripheral and central tissues.

## 3. Endotoxemia

LPS-induced systemic inflammatory toxicity is termed ‘endotoxemia’. The GI tract of mammalian species including humans is a densely vascularized structure with an intricate innervation aligned by the epithelial cell layers. The microbiota is a member of an ecological system comprising trillions of microorganisms within a defined environment. The genetic composition of microbes is termed the microbiome [19], which can be altered by many environmental factors, including various diets. The gut and brain have a bidirectional relationship, termed the gut–brain axis (GBA). Through the GBA, the pathological changes in the CNS negatively affect the enteric system and vice versa, where gut dysbiosis promotes and/or aggravates neurological disorders [20]. Among the many possible connections between the gut and brain, the vagus nerve plays a predominant role. A leaky gut is characterized by the increased permeability of the intestinal barrier allowing the translocation of various microbes and their potentially toxic metabolites into the systemic circulation. Increased relative abundance of pathogenic microorganisms and reduced levels of beneficial bacteria accompanied by altered amounts of microbial-derived metabolites is termed gut (microbial) dysbiosis. An imbalanced microbial composition causes an aberration of the epithelial and mucus layer of the GI tract by dysregulating desmosomes, adherens junction (AJ) and tight junction (TJ) proteins. Through these changes, the IgA antibodies are released from lamina propria and pro-inflammatory cytokines from macrophages and monocytes [21]. Alterations in the intestinal microbiota affect immune homeostasis through major changes in the numbers of dendrite cells, T helper cells, and B cells [22]. Further, microbial-derived metabolites such as LPS, short-chain fatty acids (SCFAs), and indoles/kynurenine can reach the CNS via the defective BBB, decrease the activity of neurons, activate microglia and astroglia, and thus play a crucial role in promoting neuroinflammation, thereby contributing to neurodegeneration [23,24]. Increased systemic levels of LPS produce symptoms like anhedonia, lethargy, sleepiness, and anxiety, known as sickness behavior [25]. Thus, it is clear that increased systemic LPS levels negatively affect brain function, and many reports indicate a strong correlation between LPS levels and various NDDs. In the present manuscript, we briefly summarized the reports linking systemic LPS with major NDDs such as AD, PD, MS, ALS, and cerebrovascular disease. We utilized biomedical search engines including PubMed, SCOPUS, ScienceDirect, and Google Scholar to survey the scientific literature.

## 4. LPS-Mediated Inflammatory Mechanisms

LPS, a cell-wall immunostimulatory component of Gram-negative bacteria, varies in terms of its ability to act as a neurotoxin primarily depending on its lipid A characteristics. LPS derived from *E. Coli* is found to cause severe inflammation compared to LPS derived from *R. sphaeroides*. LPS was foremost recognized as a Toll-like receptor 4 (TLR-4) ligand [26]. LPS interacts with the Toll-like receptors (TLRs) [27] and nucleotide-binding oligomerization domain-containing protein (NOD)-like receptors, which are classified as pattern recognition receptors (PRRs). LPS binding to TLR4 is facilitated by LPS binding protein (LBP) [28], resulting in formation of the TLR4-MD2 complex [29] via the myeloid differentiation factor 88 (MyD88)-dependent pathway. This complex promotes nuclear factor-kappa B (NF-κB) activation, which is a redox sensitive critical transcription factor for the production of many pro-inflammatory cytokines and chemokines [30]. In addition, LPS causes MyD88-independent NF-κB-mediated interferon-1 release [31] (Figure 2). 

Within the CNS, microglia represent the resident macrophages in the brain and they play a prominent role in neuroinflammation by up-regulating the immune responses [32]. TLR-4 is primarily expressed on microglia. LPS also damages the BBB, stimulates apoptosis of neuronal cells, and induces cognitive impairment and neuroinflammation by activating the microglia via the NF-κB signaling pathway. Chronic or prolonged polarization (or activation) of microglia by LPS leads to neuronal death and a surge in proinflammatory cytokines, such as tumor necrosis factor (TNF)-α, interleukin (IL)-1β, prostaglandin E2 (PGE2), and nitric oxide (NO), especially in the hippocampus [33,34]. These cytokines and toxic compounds are found to be the central mediators of neuroinflammation and neuronal damage. Specifically, LPS-related TLR-4 activation causes extensive neuronal cell death. High levels of LPS in the circulation induce hyperactivation of the microglia and astrocytes, leading to neuroinflammation with impaired neurons, synapses, and cognitive functions [35]. Several animal studies have shown that LPS administration induces cognitive impairment [36,37] and a wide range of complex psychological behaviors such as anorexia, diminished locomotion, exploratory behavior, weight loss, aggravated anxiety, somnolence, and general depression. Most of these symptoms are found to be concurrent to those clinically relevant symptoms observed in patients with NDDs. In addition, systematic administration of LPS has been shown to induce learning impairment and memory deficit, increase amyloid beta (Aβ) accumulation, and can promote synaptic and memory dysfunctions [38,39]. Consequently, LPS administration is frequently used to study neuroinflammation-associated diseases in experimental models, including rodents and cultured neuronal cells. Outcome results using LPS administration in rodents or cultured cells often vary depending on the dose, bacterial source, and treatment time of LPS as well as the use of single or multiple injection methods (a few times) and different routes (intraperitoneal or intracerebroventricular). In fact, repeated administrations of LPS are also used to evaluate changes in neuroinflammation and behaviors [40,41]. LPS injections stimulate microglia by activating the NF-κB signaling pathway, which leads to loss of neuronal cells in the hippocampus and cortex region. Elevation in neuronal cell death and Aβ levels were observed as a function of systemic and neuroinflammation post LPS injection, finally resulting in sickness behavior and cognitive impairment as reflected by low scores in the Morris water maze and other behavioral tests such as a passive avoidance test. The levels of anti-inflammatory cytokines IL-4 and IL-10 in the serum and brain were decreased via LPS treatment, whilst pro-inflammatory cytokines TNF-α, IL-1β, PGE2, and NO were increased. In addition, LPS up-regulated the protein expression of cyclooxygenase-2 (COX-2) and inducible NO synthase (iNOS) in the brain tissue probably via activation of the transcription factor NF-κB (Figure 2). Indeed, VIPER (viral inhibitory peptide for TLR4), which is a TLR-4-specific inhibitory peptide, inhibited LPS-induced neuroinflammation and cognitive impairment [42].

## 5. LPS-Mediated Oxidative/Nitrative Stress and Neurodegeneration

In addition to having a significant role in numerous neurological disorders, many previous studies showed that LPS exposure also enhances ROS and RNS accumulation, resulting in amplified production of pro-inflammatory mediators. Systemic administration of LPS induces neuroinflammation, which is known to cause harmful effects on brain function [43,44,45] such as memory impairment and neuroinflammation through TLR4/NF-κB signaling with elevated expression of inflammation-related genes and/or molecules, including various proinflammatory cytokines/chemokines. A study with both in vitro and in vivo models demonstrated that neuroinflammation and neurodegeneration are caused by LPS exposure via the activation of microglia, NF-κB, and the p38/c-Jun N-terminal Kinase (JNK) pathway. Furthermore, LPS activated phosphorylated-JNK (p-JNK, a stress activated protein kinase pathway), frequently associated with Bax- and mitochondria-dependent cell death pathway [46], which culminated in neuroinflammation and neuronal damage [47]. A recent study showed that systemic LPS administration elevated the accumulation and production of ROS/RNS and oxidative/nitrative stress with decreased glutathione levels, leading to neuroinflammation, synaptic loss, neurodegeneration, and memory impairment in the adult mouse brain [48]. LPS administration enhanced the activated p-JNK levels in the dentate gyrus (hilum and granular cells) and Cornu Ammonis 3 (CA3) (molecular layer and pyramidal cells) regions of the hippocampus in comparison to saline-treated animals. Mounting evidence shows that LPS promotes apoptotic neurodegeneration in adult mice by triggering mitochondrial apoptotic and neuroinflammatory pathways via the upregulation of several apoptotic markers such as Bax, cytochrome C release, caspase-9, and caspase-3 cleavage. Western blot analyses of the hippocampi extracts showed increased protein levels of cytochrome C, cleaved caspase-9, caspase-3, apoptotic protease activating factor 1, and poly (ADP-ribose) polymerase-1, indicating LPS-mediated neural damage in adult mice [49]. Fluoro-Jade B and Nissl staining results revealed that LPS-treated rodents showed increased numbers of damaged, shrunken, and degenerative neuronal cells in the cortex as well as CA1, CA3, and dentate gyrus regions of the hippocampus. LPS administration caused significant synaptic dysfunctions and memory impairment through down-regulation of pre-synaptic proteins synaptosomal-associated protein 23 (SNAP-23) and synaptophysin, and post-synaptic proteins PSD-95, phospho-glutamate receptor (p-GluR1), and phospho-cAMP response element-binding protein (p-CREB) in the hippocampus of adult mouse brains [48]. Importantly, another study showed that even a single systemic LPS injection could impair or decrease long-term potentiation, spatial memory, and neurogenesis in the hippocampus [50].

## 6. Effects of LPS on Models of Major Neurodegenerative Diseases

Neuroinflammation is considered an important factor in the neurodegeneration process contributing to cognitive impairment inherent to major neurodegenerative diseases, such as AD, PD, MS, ALS, and Huntington’s disease (HD) [51,52,53]. LPS- or Aβ-induced production of proinflammatory molecules or cytokines such as NO, TNF-α, and IL-1β from microglia are important hallmarks of AD, PD, MS, and cerebral ischemia or stroke [54,55,56]. Post-mortem brain samples of patients with NDDs have demonstrated evidence of significant astrogliosis, activation of microglia, and increased levels of proinflammatory cytokines [57,58]. 

### 6.1. Alzheimer’s Disease 

Alzheimer’s disease (AD) is a chronic NDD characterized by increased deposition of intercellular amyloid-beta (Aβ) plaques and intracellular tangles of misfolded phosphorylated tau-proteins. Reports indicate that neuroinflammation aggravates AD pathology [59]. LPS causes neuronal inflammation by negatively affecting BBB integrity in mice [60]. The inflammatory response in AD begins with the binding of microglial cells to the pre-existing amyloid-beta fibrils and soluble amyloid-beta oligomers through cell surface proteins such as CD36, CD47, and TLRs. The activated microglial cells produce proinflammatory cytokines such as IL1β, IL6, IL12, IL18, and TNFα. These cytokines in turn down-regulate the genes responsible for Aβ clearance, leading to impaired autophagy and further Aβ accumulation [20]. A study showed that injection of LPS up-regulated TLR4 and down-regulated TREM2 protein expression in 3X- or APPswe/PS1 AD transgenic mice. Other scientists also showed microglial hyperactivation and apoptosis of the hippocampal neurons in LPS-exposed mice [61,62]. In another study, the intranasal application of LPS in mice increased the number of long noncoding RNAs (lncRNA) in the hippocampus, demonstrating its proinflammatory role in AD [63]. Agostini et al. reported sex-dependent effects on metabolism, leading to hippocampal neuronal death after intravenous injection of LPS. The female mice showed greatly increased levels of pro-inflammatory markers compared to males, but the mechanism behind this difference in sex-dependent vulnerability is yet to be studied [64]. Some studies with APP-transgenic mice (such as APPswe/PS1 Tg mice and 4- or 12-month old 3xTg-AD) showed that systemic LPS injection triggers microglia activation, leading to neuroinflammation, Aβ accumulation and/or tau pathology, synaptic loss and neurodegeneration with learning impairments, memory deficits, and cognitive decline [65,66,67]. Other reports indicate that a disruption of synaptic function is an important primary feature of AD accompanied with memory impairments and cognitive dysfunction with or without the induction of neurodegeneration [68,69].

In AD patients, microglial overactivation is identified as an early key pathogenic event that occurs before neutrophil-mediated destruction of the BBB and neuronal cells via releasing neutrophil extracellular traps (NETs) [70]. LPS levels were increased by two-fold in the neocortex [71] and three-fold in the hippocampal regions of the brain in AD patients, indicating gut dysbiosis and elevated intestinal barrier dysfunction (i.e., leaky gut) [72]. Gram-negative *Bacteroides fragilis* produces a unique type of LPS, which has a structural variation in the lipid A region, promotes severe systemic inflammation, and is recognized by TLRs and CD14 on microglial cells [73]. Through the Toll-interacting proteins (TOLLIP), LPS stimulates activating transcription factor 2 (ATF2), causing the release of mitochondrial ROS. Chronic administration of LPS promotes prolonged activation of microglial cells with elevated nitrosative/nitrative and oxidative stress. The increased ROS and RNS participate in a neurodegenerative pathological cascade, reflecting the multi-dimensional neurotoxin effect of LPS [74]. Even at a lower concentration in the cerebrospinal fluid (CSF), LPS is capable of stimulating pro-inflammatory responses through cell surface IL-1R-associated kinase-1, which in turn impairs autophagy and causes proteinopathy in AD patients, as illustrated in Figure 3.

### 6.2. Parkinson’s Disease 

Parkinson’s disease (PD) is the second major neurodegenerative disease with impaired motor and non-motor functions in patients. PD is primely characterized by a progressive and selective destruction of the nigrostriatal dopaminergic system [75]. Locomotor impairment symptoms include bradykinesia, tremor, and rigidity, all of which are pathologically related to the loss of dopaminergic neurons in the pars compacta region of the substantia nigra of the brain. Around 80% of PD patients suffer from GI dysfunction, specifically constipation and intestinal inflammation with leaky gut, which can precede locomotor symptoms by years [76]. In PD, neuroinflammation is one of the major contributors to the development of the disease [77]. Studies using rodent models have shown that orally administered LPS can induce intestinal PD pathology [78,79]. In a PD model of experimentally induced inflammation in rats 10 weeks after the initiation of supra-nigral and continuous infusion of nanogram quantities of LPS for a period of two weeks, a 70% progressive loss of nigral dopaminergic neurons was noted. Similarly, an in vitro cell culture model of PD indicated significant production of NO, TNF-α, and superoxide, which are the primary factors involved in LPS-induced neurodegeneration. LPS-induced elevation of NO and superoxide is likely to produce potently toxic peroxynitrite, which promotes nitration of Tyr and S-nitrosylation of Cys residues of many proteins in the mitochondria and endoplasmic reticulum (ER), leading to mitochondrial dysfunction, ER stress, and cell death, as reviewed [80,81]. However, a single intranigral injection of microgram levels of LPS induced neurodegeneration within a few days [82,83]. Comparison of rat PD models induced by a single intra-nigral injection (acute) and infusion (chronic) of LPS into the substantia nigra pars compacta region showed that degeneration in both nigral dopaminergic nerve cells and γ-aminobutyric acid (GABA)-containing neurons was observed in the acute model [83], while in the chronic model [82], nigral dopaminergic neurons were selectively degenerated. Another important difference was that LPS-induced microglial activation was observed either at the same time or immediately prior to apparent neurodegeneration in the acute model. In contrast, in the chronic model, LPS-induced neurodegeneration began only weeks after the peak of microglial activation despite using similar LPS doses (~5 µg LPS). 

LPS administered into rats was shown to up-regulate cell surface receptor CD14 expression within a specific cellular population including microglial cells by binding to CD14/TLR4 complex [84]. It causes microglial activation and the release of pro-inflammatory cytokines and neurotoxic factors such as IL-1β, TNF-α, prostaglandins, superoxide, and NO that damage dopaminergic neurons [49] via mitochondrial dysfunction and apoptosis [85,86]. In an in vitro study, intestinal epithelial IEC-6 cells exposed to LPS down-regulated the expression of tight junction proteins ZO-1, occludin, and epithelial-cadherin (e-cadherin) [87]. Mice infected with Gram-negative bacteria (*P. mirabilis*) showed increased LPS levels in serum and fecal samples, suggesting elevated intestinal barrier dysfunction (i.e., gut leakiness) and endotoxemia, which can contribute to destruction of the BBB, activation of microglia, and neurodegeneration through mitochondrial dysfunction and apoptosis [85,86]. These bacterially-infected animals showed increased levels of proinflammatory cytokines and high rates of apoptosis of dopaminergic neurons [87]. Bronstein et al. showed that exposure of glial cells to LPS and 6-hydroxydopamine resulted in 89% loss of tyrosine hydroxylase (TH) immunopositivity, wherein LPS toxicity was found to be minimal. In a similar study, co-cultured neuronal-glial cells exposure to LPS produced 70% loss of TH-immunopositive neurons, clearly indicating that LPS mediates dopaminergic cell death through participation of glia cells [88]. LPS (10 µg/mL administered in drinking water for 12 days) caused motor dysfunction assessed via the tail suspension test in Thy1-αSyn over-expressed mice compared to the corresponding normal wild-type mice [89]. In addition, altered distribution of TJ ZO-1 and AJ protein e-cadherin was observed when intestinal epithelial IEC-6 cells were exposed to LPS, strongly suggesting changes in epithelial cell permeability [89]. 

The GI tract is the first organ affected in PD patients, as they show abnormal staining for the *Escherichia coli* and α-synuclein in their colon. In PD, the hyperpermeability of the intestinal mucosa correlates positively with elevated intestinal staining for *E. coli*, 3-nitrotyrosine (a marker of nitrated proteins), and α-synuclein. Metagenome analysis or 16S rRNA amplicon sequencing showed increased relative abundance of Lactobacillus, whilst the absolute counts of the *Clostridium coccoides* group, the *Bacteroides fragilis* group, and the putative hydrogen-producing bacteria Prevotella [90,91] were decreased in PD subjects compared to those of healthy controls [92]. In PD subjects, the 16S RNA sequencing analyses revealed significant increases in LPS-producing Gammaproteobacteria and mucin-degrading Verrucomicrobiae with non-significant decrements in bacteroidia and clostridia when compared to healthy subjects [89]. Analysis of fecal microbiome showed significant reduction of several metabolism-related genes in PD patients, while expression of the genes associated with LPS biosynthesis and type III bacterial secretion systems were significantly enhanced in PD patients [90]. To understand the gut microflora of PD patients, 16S RNA sequencing of the volunteers was performed and a higher population of Gram-negative bacteria were detected, which was found to be the central reason for inflammatory process in the intestine and an important triggering factor for the misfolding of alpha-synuclein and accumulation of Lewy bodies [93], as summarized in Figure 4.

### 6.3. Multiple Sclerosis

Multiple sclerosis (MS), a chronic autoimmune neuroinflammatory disease, is characterized by the demyelination of neurons, gliosis, and various degrees of axonal and oligodendrocyte pathology. It is an immune-mediated disease influenced by both genetic and environmental factors [94]. It causes a wide array of neurological deficits, such as loss of vision, numbness, fatigue, and cognitive and motor impairment that vary depending on the location of the lesion [95]. Activated lymphocytes (such as CD4+ Th1, Th17, CD8, T cells, and monocytes) and macrophages enter the CNS through areas of inflammation called perivascular cuffs in MS and the meningeal barrier to negatively affect the function of parenchyma cells of the brain. Eventually, infiltration of these activated immune cells results in demyelination, gliosis, and disruption of neuronal signals via neuroaxonal damage [96]. The notion of increased levels of LPS-binding protein (LBP, a gut leakiness biomarker), which can trigger spontaneous autoimmunity in mouse brain along with the presence of abundant immune cells (such as T lymphocytes) in the CNS of MS patients, suggests that MS is an immune-mediated neuronal disorder. An intracerebral injection of LPS induces BBB damage by promoting the ingress of immune cells (leukocytes) with the presence of major histocompatibility (MHC) class II antigen and an increased T lymphocyte count, which accounts for damage to the myelin sheath and parenchymal cells of the rat brain. LPS-induced MS models showed higher levels of LPS and LBP in the brain, spinal cord, and blood of rats, in similarity to that seen in MS patients [97], indicating the crucial role of gut microbial dysbiosis and leakiness in MS pathophysiology [98]. Metagenomic signatures of fecal microbiomes reported an increased Firmicutes/Bacteroidetes ratio, higher Streptococcus and reduced Prevotella strains in MS patients. Other researchers found high counts of *Akkermansia muciniphila* and *Acinetobacter calcoaceticus* with low abundance of *Parabacteroides distasonis* in patients compared to healthy individuals [99]. In MS pathophysiology, intestinal permeability changes are demonstrated by low-grade microbial translocation into systemic circulation and eventually to the brain along with a low-grade endotoxemia in MS patients [100,101]. These results show that increased levels of LPS and LPS-binding protein in the plasma of MS patients correlate positively with the higher concentrations of pro-inflammatory cytokines and the expanded disability status scale (EDSS). In addition, LPS exerts its pro-inflammatory actions on microglia and astrocytes, leading to disruption of the BBB, which subsequently can perpetuate the pathogenic loop of MS. Hyperactivation of microglia induces neuroinflammation by secreting pro-inflammatory cytokines such as TNF-α, IL-1β, and IL-6 along with increased levels of NO, ROS, iNOS, and COX-2 (biomarkers for oxidative stress), all of which can lead to myelin sheath aberration. LPS-induced hyperactivation of microglial cells stimulates NF-κB and phosphatidylinositol 3-kinase/Akt signaling pathways, which are known to cause neuroinflammation, neurodegeneration, and MS-like pathology [102]. 

In MS patients, gut dysbiosis-mediated increase in plasma LPS leads to the dysfunction of the mucosal barrier and destruction of the BBB, neurons, and myelin sheath [97]. Aberrations in immune regulation marked by defective regulatory T cells and activated aryl hydrocarbon receptors and pro-inflammatory T helper (Th) 17 cells stimulate excessive cytokine release and inflammation in MS. Recent studies have shown the important function of ROS/RNS in MS pathogenesis by activating microglial cells [103]. Increased levels of ROS/RNS can stimulate demyelination and mitochondrial dysfunction, leading to neuronal cell death, thus resulting in MS pathology [104]. In fact, MS patients were reported to have a deficiency of global antioxidants and a surge in oxidative stress biomarkers [105]. 

Pentraxin, another acute-phase protein, which is a pro-inflammatory biomarker of inflammatory diseases, was found in higher levels in the CSF of MS patients [106]. Injection of LPS into rat’s dorsal funiculus induced focal inflammation, reflected by an increased number of polymorphonuclear cells, leading to demyelination and lesions in the injected area [6,84]. Acute-phase proteins such as chemokines released from platelets, macrophages, and endothelial cells are found to be activated by LPS. In particular, CCL5 induces the migration and recruitment of mast cells, T cells, basophils, and natural killer cells, resulting in chronic inflammation. Other cytokines like serum amyloid proteins, which are produced in the liver, were elevated by LPS administration [107]. LPS injection into mice can also stimulate significant neuronal loss [5], which concurs with neuronal damage observed in the hippocampus of MS patients [108].

### 6.4. Amyotrophic Lateral Sclerosis

Amyotrophic lateral sclerosis (ALS) is a fatal progressive neurodegenerative and neuromuscular disorder affecting the brain and spinal cord [109,110]. It is characterized by progressive loss of upper and lower motor neurons in the motor cortex, brainstem, and spinal cord [111], which leads to muscular atrophy, wasting, weakness, progressive paralysis, and respiratory failure that eventually results in the death of affected individuals. The pathogenesis of ALS is known to be familial in approximately 10% of cases, and in 20% of cases, ALS is caused by mutation in the genes coding for superoxide dismutase-1 (SOD1) [112]. In ALS pathophysiology, LPS acts as a potent neuroinflammatory toxin and immunotoxin, stimulating CD14 receptor expressing cells [28] such as monocytes, granulocytes, and macrophages, polarizing the microglia and triggering the release of pro-inflammatory markers such as TNF-α and IL-6. Binding of LPS to TLR4 of the immune cells further aggravates the pro-inflammatory loop [113], leading to increased oxidative stress, neuroinflammation, and neurodegeneration. Particularly, injection of LPS into SOD1 (G37R) mutant ALS mice induced a significant reduction in their lifespan, indicating that LPS-mediated macrophage activation can intensify the pathogenesis of ALS in vivo. LPS induces classical monocyte activation and production of pro-inflammatory cytokines like IL-6 and TNF-α [114]. ALS patients were found to have increased levels of monocyte chemotactic protein-1 (MCP-1) and IL-6 in the cerebral spinal fluid (CSF) and sera, while levels of TNF-α were elevated in the blood. The blood samples of sporadic ALS subjects also had elevated levels of abnormally activated monocytes or macrophages compared to control patients, suggesting the high inflammatory response to endogenous LPS [115]. A study revealed the presence of increased levels of endotoxin/LPS in plasma from subjects with sporadic ALS and AD when compared to healthy controls. Elevated LPS levels correlated positively with the intensity of blood monocyte/macrophage activation in MS patients and were found to be significantly elevated in patients with advanced ALS [116]. Both systemic LPS levels and LPS-activated monocyte/macrophage are considered as two new important co-factors that might play substantial roles in ALS pathogenesis and thus may represent novel targets for therapeutic intervention in ALS. Overall, the pathological findings in ALS subjects were coherent with preclinical studies based on marked inflammation with activation of astrocytes and microglial cells, increased oxidative stress, and induction of expression of pro-inflammatory enzymes and cytokines. 

Increased LPS levels cause chronic low-grade inflammation, which is one of the prime contributing factors for neurodegeneration in ALS (and possibly other NDDs). Both ALS animal models and patients have shown infiltration of T cells, peripheral proinflammatory monocytes, and dendritic cells (DCs), indicating the vital role of inflammatory cells in disease progression. Both CD4+ and CD8+ T lymphocytes infiltrate the brains of ALS subjects and interact with glial cells (microglia and astrocytes) contributing to motor neuron degeneration. ALS subjects showed a significant reduction in the number of circulating DCs. The DCs regulate the activation of T cells [117] via presenting peptide MHC complex, expressing co-stimulatory molecules, and release of pro-inflammatory cytokines. The degree of immune response elicited by DCs depends upon the intensity of the signal it perceives [118]. DCs from ALS patients had an elevated expression of CD62L and a predisposition to overexpress CCR2 in comparison with healthy donors. DCs obtained from a subpopulation of ALS patients secreted increased levels of IL-8 and CCL-2 upon LPS stimulation [119]. 

DNA micro-array analysis of post-mortem spinal cord tissues from sporadic ALS subjects revealed changes in the expression of the genes associated with the inflammatory process. Consistently, autopsy of sporadic ALS patients and analyses of mutant SOD1 transgenic mice have demonstrated the involvement of both innate and adaptive immunity in neurodegeneration. Genetically targeted deletion of mutant SOD1 proteins in astrocytes and microglia significantly increased their survival, proving the predominant role of glial activation and secondary neurotoxicity in the mutant SOD1 model. Recent findings have provided substantial evidence on the toxic functions of mutant SOD1 proteins accompanied with activated NADPH oxidase (NOX) activity and elevated ROS production. Further evidence suggests that the upregulation of NOX correlates positively with gut dysbiosis in ALS patients [120]. A clinical genetic study investigating the changes in inflammation-related gene expression showed increased MyD88-independent LPS/TLR4 signaling and up-regulation of the expression of G-protein coupled receptor-43 in ALS patients in comparison to control subjects. The increased levels of these genes were linked to peripheral blood monocyte activation [121]. C/EBPβ, a transcription factor, was found to be up-regulated in microglia cells in the spinal cord of ALS subjects, explaining the exacerbated neuroinflammation [122]. 

### 6.5. Stroke 

Stroke is the second leading cause of global mortality and a leading cause of acquired disabilities with a global lifetime risk of 25% [123]. Stroke is caused by interruption in the supply of blood due to blockage or rupture in the cerebral blood vessels, leading to increased intracellular calcium and excitotoxicity. Cerebral ischemia deprives neuronal cells of their oxygen and essential nutrients, resulting in neuronal cell death and brain damage. Several studies have shown that gut microbial dysbiosis can be considered a potential risk factor for stroke development and post-stroke prognosis [10,124,125]. Multiple experimental studies in stroke-related animal models have revealed a direct association between gut microbial dysbiosis and inflammation, gut motility, intestinal permeability change, and altered immune response through dysregulated GBA signaling. Similarly, the pathophysiological mechanisms underlying gut dysbiosis in stroke patients report collapse of intestinal epithelial barrier TJ and AJ proteins, altered mucus secretion, gut dysmotility, loss of goblet cells, and alterations in the local immune homeostasis, leading to systemic inflammation with elevated endotoxin levels [126]. These changes in turn alter intestinal and systemic immune homeostasis, leading to poor stroke prognosis. 

Subsequent endotoxemia contributes to increased systemic inflammation, neuroinflammation, disruption of BBB, and neurotoxicity [127], accompanied by elevated levels of ROS, RNS, trimethyl amine oxide, and NOX 2/4. However, the levels of total antioxidant capacity, including the reduced glutathione (GSH), and SOD activity were reported to be reduced in brain tissues of the stroke mice [128,129,130]. Mechanistic studies in experimental stroke mice demonstrated that increased intestinal permeability caused increased translocation of LPS and gut microbes to the brain and negatively affected its function through the GBA. A study on C57BL/6J mice demonstrated that exposure to LPS prior to transient middle cerebral artery occlusion (MCAO) worsened the infarct size and neurological deficits with reduced cognitive ability and locomotor activity [131,132].

The inflammatory reactions with higher infarct size, synapse loss, and glial dysfunction are marked by high levels of LPS, C-reactive proteins, TLR4, bacterial toxins, toxic metabolites, and pro-inflammatory cytokines. These damaging changes cause the intensification of neuronal deficits and account for poor neurobehavioral outcomes. Inflammatory signals from the infarct site further enhance oxidative stress, which promotes glial and neuronal cell senescence, leading to further production of damage-associated molecular patterns (DAMPs) that bind to pattern recognition receptors, further aggravating the pro-inflammatory loop. DAMPs and ROS are also responsible for the weakening of the BBB in stroke patients. BBB dysfunction allows LPS and other gut-derived toxic molecules to penetrate and damage neuromodulator activity in stroke patients. Bacteria residing in intestines have been found to be a major cause of post-stroke infections in both patients and animal models [131]. Post-stroke patients are prone to infections like bacteremia and pneumonia in which the systemic levels of pro-inflammatory cytokines were elevated [133]. It was also observed that patients with *Helicobacter pylori* bacterial infection are at higher risk of developing an ischemic stroke due to the presence of increased LPS levels [134]. An NIH Stroke Scale (NIHSS) study investigating the correlation between LPS levels and stroke outcome found that higher LPS levels were positively associated with worse stroke outcome. Evidence suggests that increased neuronal death and epithelial layer necrosis were observed in stroke patients due to inflammation triggered by LPS [135]. 

## 7. Summary

The pathogenic role of gut dysbiosis in NDDs is gaining greater attention due to the growing interest in the bidirectional signaling pathways between the gut and the brain through the GBA. Both animal models and clinical studies have shown that gut microbial dysbiosis, which is usually accompanied by enhanced intestinal permeability, allows the translocation of pathogenic bacteria and gut microbial-derived toxic metabolites into the systemic circulation, thereby providing access to other organs including the brain. Specifically, LPS, an important cell wall component of Gram-negative bacteria, is considered an endotoxin and a neurotoxin, as it activates the immune cells both in the periphery and brain, leading to chronic local and systemic inflammation, including neuroinflammation. LPS has the potential to stimulate BBB damage and neuronal cell death through its lipid A component. Neuroinflammation is a key component in neurodegeneration, which leads to neuronal loss, synaptic dysfunctions, learning and memory impairments, and motor skill decline, classic symptoms frequently observed in many NDDs in both patients and experimental animal models. LPS increases the levels of pro-inflammatory cytokines, which in turn disrupt the BBB and thus allow excessive bacteria and LPS or other toxic bacterial products to enter the brain parenchyma and elicit chronic neuroinflammation and neuronal cell death, contributing to NDDs. The brain-resident immune cells such as astrocytes, DCs, and microglia also play an important role in inflammatory processes. The receptors that mediate LPS-induced inflammation are NOD, TLR4, CD14, and LPS binding proteins. Clinical and preclinical data strongly suggest that as a result of aggravated acute systemic inflammation, neuroinflammation, and neuronal damage, LPS is capable of predisposing, inducing, and propagating various NDDs. 

In conclusion, these findings undoubtedly underscore the important pathogenic role played by LPS in mediating systemic and neuroinflammatory processes in various NDDs. In the future, new therapeutics targeting LPS reduction and/or LPS-mediated inflammation should be actively explored as promising neuroprotective candidates for the treatment of NDDs. Further mechanistic studies are warranted to clearly understand the interactions between LBP and LPS, inflammation, and oxidative stress in neurodegeneration.

## Figures and Tables

**Figure 1 cells-11-04038-f001:**
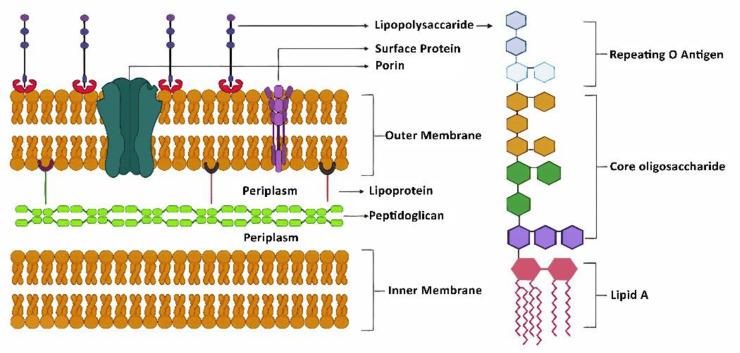
Structure of LPS embedded in the cell membrane of Gram-negative bacteria.

**Figure 2 cells-11-04038-f002:**
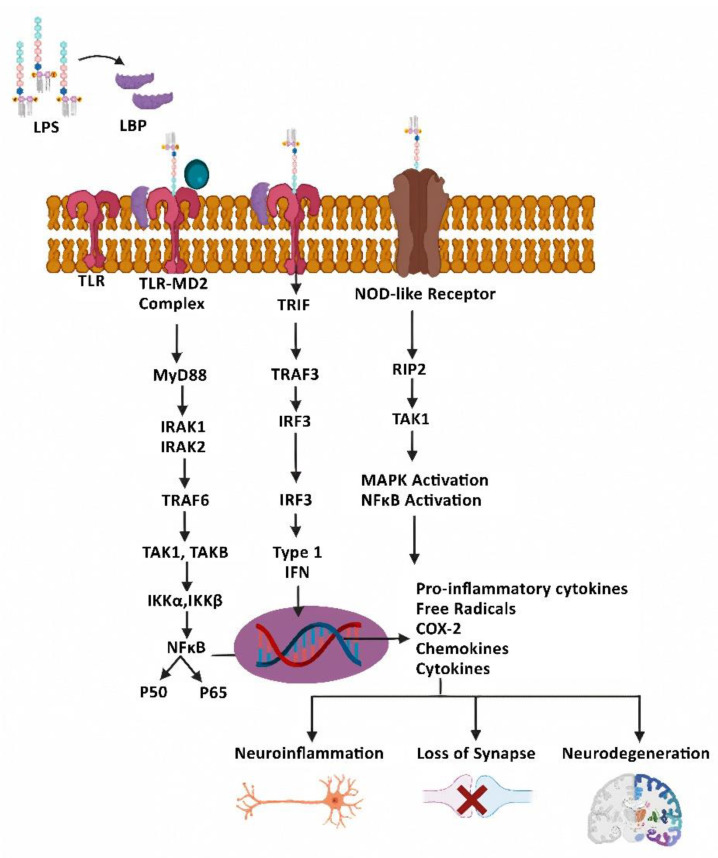
Mechanisms of lipopolysaccharide (LPS-induced neurodegeneration). LPS, with the help of lipopolysaccharide binding (LBP), binds to a different family of receptors viz NOD-like receptors and toll-like receptors (TLRs) to generate TLR-MD2 complex. The downstream signaling cascades include increased production of proinflammatory cytokines, free radicals, and chemokines, all of which contribute to synaptic loss, neuroinflammation, and neurodegeneration.

**Figure 3 cells-11-04038-f003:**
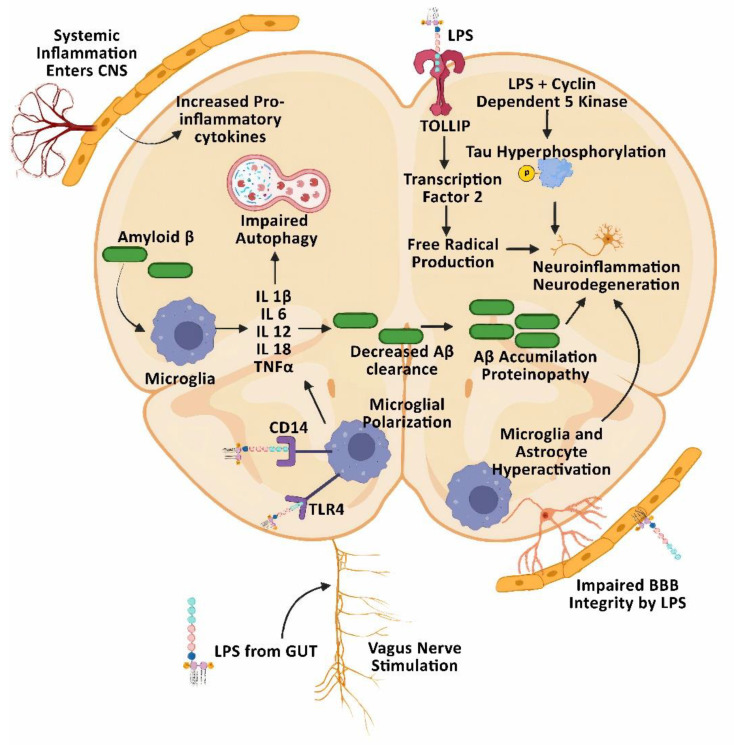
Lipopolysaccharide (LPS) induces neuroinflammation in Alzheimer’s disease (AD) brains. LPS enters the brain through the vagal nerve and systemic circulation and impairs the BBB further by binding to CD14 and TLR4 receptors. This causes microglial polarization, which further increases free radical stress, defective autophagy with elevated proteinopathy, and aggravated neuroinflammation accompanied by neuronal cell death in AD.

**Figure 4 cells-11-04038-f004:**
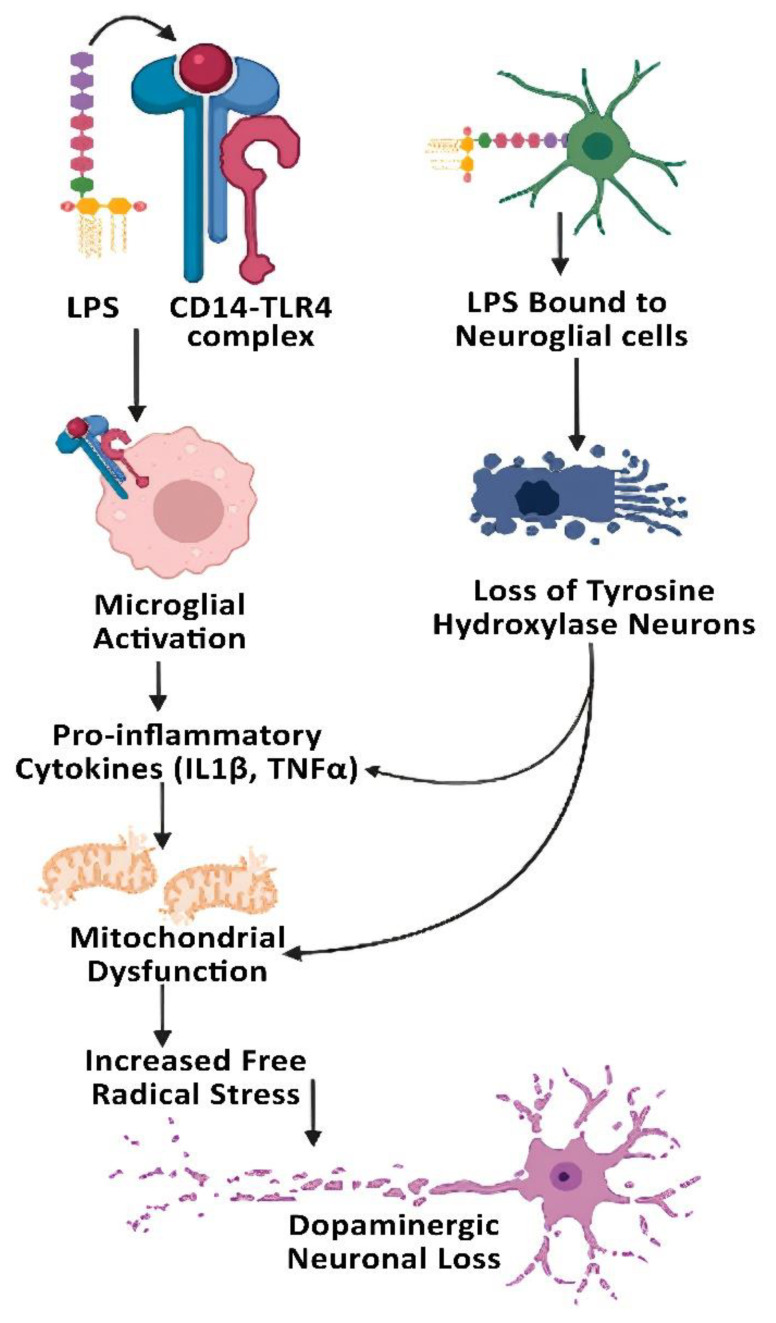
Lipopolysaccharide (LPS)-induced endotoxemia causes dopaminergic neuronal death. LPS binds to CD14 and potently activates microglial cells. In addition, LPS binds to neuronal cells and down-regulates tyrosine hydroxylase gene expression. Both pathways stimulate mitochondrial dysfunction with increased free radical production, leading to dopaminergic neuronal cell loss.

## Data Availability

The data that support the findings of this study are available in standard research databases such as PubMed, Science Direct, or Google Scholar, and/or on public domains that can be searched with either key words or DOI numbers.

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
