# Peer review of "Role of Endogenous Lipopolysaccharides in Neurological Disorders"

_cells, 2022, doi:10.3390/cells11244038_

Round 1

Reviewer 1 Report

The authors present an interesting review about the association between Endogenous Lipopolysaccharides(LPS) and Neurological Disorders. As the “Gut dysbiosis-LPS-neuroinflammation-NDDsmechanism is a hotspot among the Neurological disease, this review provides information helping the understanding of the potential pathogenesis between LPS and NDDs, point out the current research progress and the solutions that still need to be solved. The manuscript were well written. However, there are some important issues to be carefully concerned, and the acceptance of the manuscript should be delayed until the issues to be resolved properly.

Q1: Page 5: line 212, the authors are advised to cite the original reference.

Q2:The review mainly stacked the results of relevant studies and did not conduct an evidence-based review and analysis

Q3:This review is similar to the published review PMID: 36080253, please explain the differences

Reviewer 2 Report

The authors focused on the effect of endogenous LPS on neuroinflammation and neuronal cell death pathways in neurological disorders. The flow and explanation of the review are quite satisfactory. The manuscript is understandable. When the other current literature focusing on the same subject is reviewed, it is noteworthy that there are more specific reviews that describe the effect of LPS and related inflammatory pathways in only one or two neurological disorders. The manuscript contains up-to-date information and describes the most common neurological disorders in society separately.  It only drew my attention that the names of microorganisms are not written in italics. This needs to be fixed at the proof stage. I believe that the present review would be beneficial for the researchers in the field and it is suitable for publication in the "Cells" journal. 

Reviewer 3 Report

This review by Kalyan et al. sheds light on the involvement of LPS in neurodegenerative diseases. The review is thorough and well written, with individual sections described extensively and clearly. The authors have summarized the published research in a concise way which will be a great help to all readers, in general.

The review, although very informative, falls short on a few fronts (below). If the authors address the shortcomings proofreading and inclusion/correction of the pints below, I believe this work is of quality and appropriate for publication.

1.     The scientific names need to be italicized throughout the manuscript.

2.     There are a lot of statements that are not cited to the original research work. The complete absence of references throughout the text makes it very difficult for the reader to assess if it is a suggestion/indication or a fact. Some instances: introduction; especially lines 82-90, 95-100. Lines 116-118-120-124, and so on and so forth.

3.     Quality of text/labels in figures needs to be improved. They appear pixelated in current form.
